# Force-triggered rapid microstructure growth on hydrogel surface for on-demand functions

Qifeng Mu[1], Kunpeng Cui[2], Zhi Jian Wang [1], Takahiro Matsuda [3], Wei Cui [3], Hinako Kato[1], Shotaro Namiki[1], Tomoko Yamazaki[1], Martin Frauenlob [1], Takayuki Nonoyama [3], Masumi Tsuda [2,4], Shinya Tanaka [2,4], Tasuku Nakajima [2,3] ✉ & Jian Ping Gong [2,3] ✉

Living organisms share the ability to grow various microstructures on their surface to achieve functions. Here we present a force stamp method to grow microstructures on the surface of hydrogels based on a force-triggered polymerisation mechanism of double-network hydrogels. This method allows fast spatial modulation of the morphology and chemistry of the hydrogel surface within seconds for on-demand functions. We demonstrate the oriented growth of cells and directional transportation of water droplets on the engineered hydrogel surfaces. This force-triggered method to chemically engineer the hydrogel surfaces provides a new tool in addition to the conventional methods using light or heat, and will promote the wide application of hydrogels in various fields.

In nature, many specific functions of creatures are achieved by their surface microstructures. For example, the cactus can efficiently collect and directionally transport water by the rough spines on their surface[1–3]. Another example is the cephalopods that display dynamic skin patterns in response to stimuli for communication and camouflage[4–6]. These microstructures on creature surfaces are usually formed via a surface growth mechanism, which provides us with elegant paradigms for designing new biomimetic materials with specific functions and promising applications.

Recently, several attempts have been made to grow microstructures on polymer surfaces by light irradiation[7,8]. Using force as an alternative trigger to grow and chemically remodel the surface of hydrogels could be a biomimetic approach. Potentially, using force as a trigger could be simple, clean, and energy-saving in comparison to those using heat or light.

Many molecular mechanisms for force-triggered chemical reactions have been developed[9–14]. For example, the mechanical force could preferentially break a weak bond incorporated on a polymer chain and generates mechanoradicals at the broken ends, and these radicals can initiate polymerisation of surrounding monomers[13,14]. However, applying such molecular mechanisms to bulk hydrogel materials is usually challenging. This is because a small amount of bond breaking causes the catastrophic failure of a conventional hydrogel[15]. As a result, such a force-triggered chemical reaction is difficult to control and could not straightforwardly be applied for the on-demand surface microstructure growth.

Recently, we have shown that double-network (DN) hydrogels are excellent materials to bridge the above-mentioned gap between molecular mechanisms and material functions. In a DN hydrogel, the force-triggered chemical reaction could be well-controlled by macroscopic force, and the reaction could be highly efficient. The stress concentration of the rigid and brittle network caused by bond breaking could be effectively suppressed by the presence of the soft and stretchable networks[16,17]. Therefore, the force-activated bond breaking in a DN hydrogel does not cause catastrophic failure of the material, and the amount of bond breaking inside a DN hydrogel increases with

[1]Graduate School of Life Science, Hokkaido University, N21W11, Kita-ku, Sapporo 001-0021, Japan. [2]Institute for Chemical Reaction Design and Discovery (WPI-ICReDD), Hokkaido University, N21W10, Kita-ku, Sapporo 001-0021, Japan. [3]Faculty of Advanced Life Science, Hokkaido University, N21W11, Kita-ku, Sapporo 001-0021, Japan. [4]Department of Cancer Pathology, Faculty of Medicine, Hokkaido University, N15W7, Kita-ku, Sapporo 060-8638, Japan. ✉e-mail: tasuku@sci.hokudai.ac.jp; gong@sci.hokudai.ac.jp

the stress/strain imposed on the material[18]. We have shown that the concentration of mechanoradicals generated inside a DN hydrogel by bond breaking is high enough to initiate polymerisation of monomers in the hydrogel bulk and improve the mechanical performance of a DN hydrogel[19].

Applying this DN concept to engineering the surface of hydrogels is a straightforward idea. However, when applying this concept to the surface, specific problems arise. The surface structure of a DN hydrogel could be very different from bulk and depends on the templates on which the hydrogels are synthesised[20,21]. In particular, the topmost surface of typical DN hydrogels synthesised with glass substrates is covered by a layer of the soft second network that is ~1–10 μm thick[22]. Therefore, there is no double-network effect in such a surface layer. To apply the double-network concept for surface chemical modification, we need to start with hydrogels with a well-controlled surface double-network structure.

In this work, we realise the stress/strain-controlled bond breaking on the surface of DN hydrogels, which guarantees spatially controllable structure remodelling on the surface of DN hydrogels. Taking advantage of fast radical polymerisation for diverse kinds of monomers, we achieve force-triggered fast microstructure growth of diverse morphology and chemistry on the hydrogel surface for on-demand applications, such as oriented cell growth and directional water transportation.

## Results

### Strategy for force-triggered surface microstructure growth

Our mechanochemical strategy of force-triggered microstructure growth is depicted in Fig. 1a. A DN hydrogel comprises two interpenetrating networks with contrasting structures and mechanical properties. The first network, highly crosslinked and pre-stretched, acts as a rigid but brittle skeleton on the molecular scale. The second network, sparsely crosslinked and relatively concentrated, is soft and stretchable[23–25]. Owing to such contrasting structures, many first network strands break during deformation without causing catastrophic failure of the hydrogel, because the second network carries load once the strands in the first network break, which redistributes stress to other first network strands around the broken strands[26]. The fracture of the strands in the first brittle network typically occurs because of the homolytic scission of covalent bonds, which generates mechanoradicals at the broken ends of polymer strands[27,28]. The internal fracture of the brittle network is spatially controllable with location and degree of macroscopic deformation. The concentration of mechanoradicals is sufficiently high to initiate polymerisation inside DN hydrogels fed with monomers, which enables force-triggered growth of DN hydrogels in the presence of reactive monomers that act as growth-building elements, analogous to nutrients in the natural growth of living organisms.

In this work, we adopted a conventional DN hydrogel whose first network was poly(2-acrylamido-2-methylpropanesulfonic acid sodium salt) (PNaAMPS) and whose second network was polyacrylamide (PAAm). Both networks were crosslinked by N,N′-methylenebisacrylamide (MBA), respectively. Density functional theory (DFT) simulation suggested two possible positions of C–C bond breakage on the brittle network, one is at the cross-linking point, and the other is on the main chain[29].

Formation of surface microstructures on DN hydrogels through force-triggered radical polymerisation requires the DN hydrogels with a double-network structure on the topmost surface layer. It has been shown that DN hydrogels comprising an anionic first network prepared on glass moulds are covered by a nonionic second network layer that is several micrometres thick because an electric double layer is formed at the interface between the anionic first network and glass that is also negatively charged in water during the preparation of the second network[22]. To prevent the formation of the electric double layer, a hydrophobic mould was used in the synthesis of the second network for the current purpose (Supplementary Fig. 1). The surface double-network structure of the sample thus prepared was studied chemically and mechanically. The chemical signal of the first network on the DN hydrogel surface was revealed by attenuated total reflectance Fourier-transform infra-red spectroscopy (Supplementary Fig. 2). The highly crosslinked and pre-stretched first network of the DN hydrogel surface determines its elastic modulus of surface (Supplementary Fig. 3 and Supplementary Table 1). These data imply that when the second network was prepared using a hydrophobic mould, the double-network structure was constructed also on the topmost surface of the DN hydrogel without the formation of a soft surface layer from the second network. The topmost surface of DN hydrogels thus prepared was smooth with a mean roughness ($R_a$) of ~0.9 μm (Supplementary Fig. 4).

### Fast force-triggered polymerisation

The force-triggered polymerisation in a DN hydrogel immersed in a monomer solution is fast. To observe the fast radical polymerisation with the naked eye, we immersed the DN hydrogel in a concentrated N-isopropylacrylamide (NIPAm) aqueous solution and pressed it with a macro-size indenter of 4 mm-diameter. Upon pressing the indenter, the transparent hydrogel rapidly turns to turbid in the pressed region within seconds (Fig. 1b, Supplementary Movie 1). In contrast, the DN hydrogel immersed in water maintains the transparent state after indentation (Fig. 1c). The results of Fig. 1b imply that force-triggered polymer strand scission induces rapid radical polymerisation of NIPAm monomer to form poly(N-isopropylacrylamide) (PNIPAm). To further confirm the formation of PNIPAm that becomes hydrophobic above 32 °C, we indented a rod-like DN hydrogel at equal intervals and then used a fluorescent molecular probe, 8-anilino-1-naphthalenesulfonic acid (ANS) to visualize the hydrophobic PNIPAm regions at high temperature (Fig.1d, Supplementary Fig. 5)[30,31]. A bamboo-knot-like fluorescent pattern was observed, confirming the formation of PNIPAm at the indented regions.

Time-resolved near-infra-red spectroscopy reveals that the force-triggered radical polymerisation of NIPAm was almost completed within seconds (Supplementary Fig. 6a), consistent with the fast transparency change in Fig. 1b. As another example, the fast polymerisation of 2-acrylamido-2-methylpropanesulfonic acid sodium salt (NaAMPS) in a force-triggered DN hydrogel was also observed (Supplementary Fig. 6b). These results demonstrate that new polymers could be chemically tethered to the broken end of the brittle network at a very high reaction rate.

### Regioselective internal rupture on the hydrogel surface

To verify that micro-indentation induces internal fracture of the first network near the surface layer, we performed cyclic indentation on hydrogels using cylindrical micro-indenters in the air (Supplementary Fig. 7a, b). In contrast to a pure elastic polyacrylamide (PAAm) single-network (SN) hydrogel that exhibits negligible mechanical hysteresis upon cyclic indentation (Supplementary Fig. 7c), the DN hydrogel surface shows obvious hysteresis, indicating energy dissipation by the rupture of covalent bonds in the first network (Supplementary Fig. 7d). The process zones on the surface of DN hydrogel could be clearly visualized by optical images, and macro-cracks are observed at large indentation depth $L_{max}$ for small indenter (Supplementary Fig. 8). In the later study, we only limit our discussion to the case without macro-cracks. Since the amount of ruptured first network strands is roughly proportional to the mechanical hysteresis energy $U_{hys}$ between loading and unloading[19], we quantified $U_{hys}$ as a function of the indentation depth $L_{max}$ of micro-indenter on the DN hydrogel surface (Supplementary Fig. 9a). $L_{max}$ was estimated from the maximum indenter displacement at loading. As a typical example, the $L_{max}$ dependence of $U_{hys}$ for a micro-indenter of $d = 568$ μm on a bulk DN hydrogel of

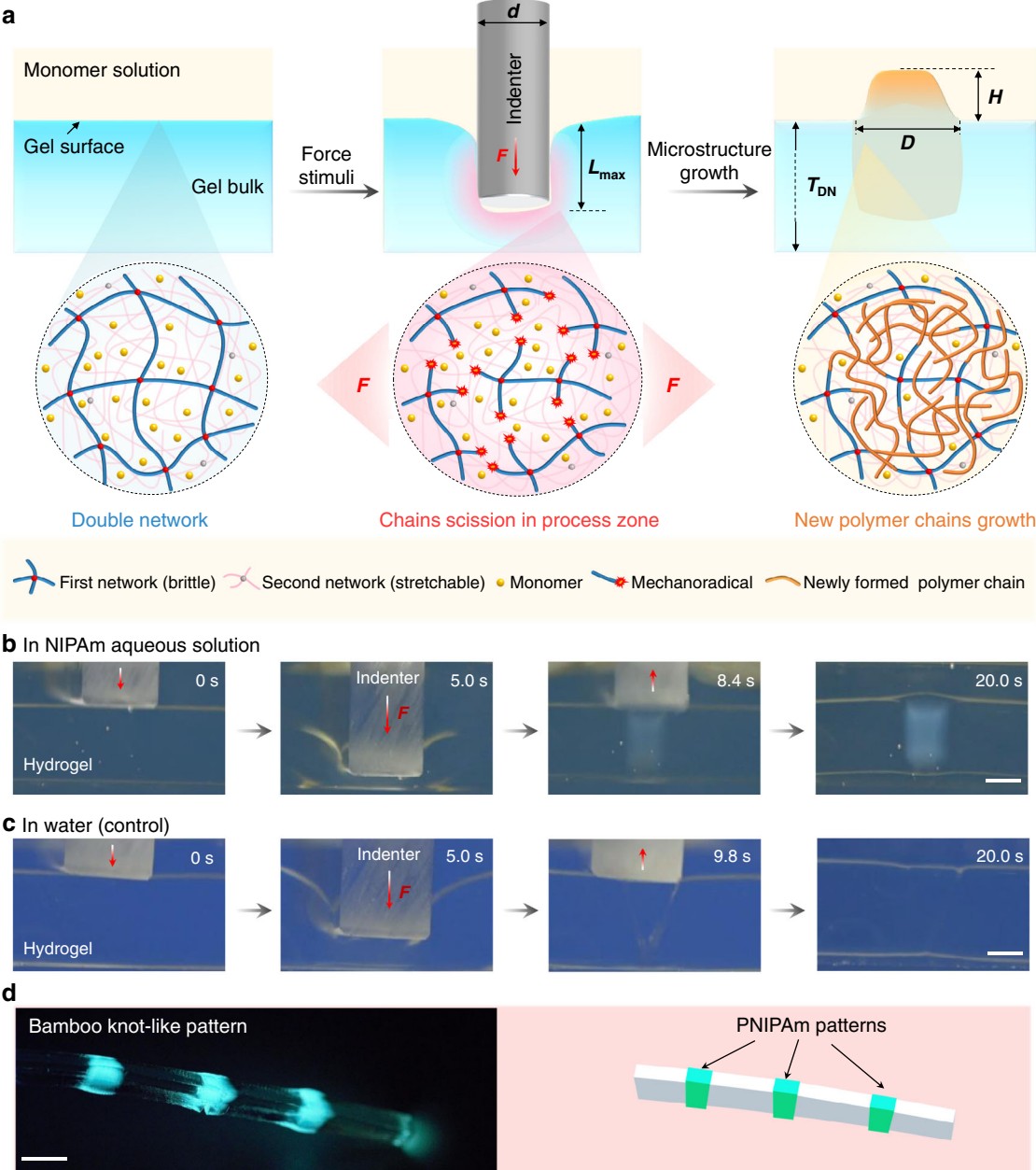

**Fig. 1 | A mechanochemical strategy to rapidly grow microstructure on DN hydrogel surface. a** Schematic illustration for rapid microstructure growth triggered by mechanical indentation under sufficient monomer supply. Molecular mechanism: the first rigid network is locally broken by compression from the cylindrical micro-indenter, which generates mechanoradicals at the broken ends of the polymer strands in the indented zone (pink region); with the presence of monomers, the mechanoradicals initiate fast growth of new polymer chains; the new polymer causes excess swelling, resulting in the formation of topological and chemical microstructures on hydrogel surface. **b** Sequential snapshots to show the rapid, regioselective force-triggered polymerisation of a DN hydrogel in NIPAm solution (monomer concentration was 1.5 M). **c** Sequential snapshots to show a DN hydrogel indented in deionized water without new polymer formation. The regioselective force in **b**, **c** was applied by a macroscale indenter to visualize the process by the naked eye. Scale bars, 2 mm. **d** Fluorescence photo to show the bamboo-knot-like structure formed by force-triggered growth of PNIPAm in DN hydrogel (Inset, cartoon representation). Scale bar, 5 mm.

3.8 mm-thick is shown in Supplementary Fig. 9b, where $d$ is the diameter of the indenter. At small $L_{max}$ (100–500 μm), $U_{hys}$ is linearly proportional to $L_{max}$, with a slope of $4.6 \times 10^{-8}$ J μm$^{-1}$. At large $L_{max}$ (500–1000 μm), the DN hydrogel surfaces undergo large and complex deformation, and the energy dissipation $U_{hys}$ increases rapidly with $L_{max}$, with a slope of about 10 times high. Beyond 1000 μm, surface micro-cracks are observed (Supplementary Fig. 7e). Assuming a process zone volume of $\sim\pi(d/2)^2 L_{max}$, the roughly estimated mechanoradical concentration increases from 0 to $2.9 \times 10^{-5}$ M as $L_{max}$ increases from 0 to 1000 μm (Supplementary Fig. 9c). These results indicate that micro-indentation to the DN hydrogel surface induces

internal bond rupture of the first network, as like that observed for homogeneous tensile deformation of a bulk DN hydrogel. At an indentation depth of $L_{max} = 1000$ μm, the mechanoradical concentration is as high as ~$10^{-5}$ M, comparable to bulk deformation at a deformation beyond the yielding point and should be high enough for surface microstructure growth[19].

## Regioselective microstructure growth

To realise surface microstructure growth, DN hydrogels were immersed in aqueous solutions of NIPAm monomers. The force-triggered polymerisation was carried out in an argon atmosphere

using a homemade experimental setup. For the DN hydrogels indented in monomer solutions and then washed in pure water, a large increase of height at the indented position is clearly observed (Fig. 2a). As DN hydrogels indented in air and immersed in water show a much smaller height change (Supplementary Fig. 10), the observed large height increase is attributed to the excessive swelling of the indented zone by the growth of new polymers tethered to the brittle network. After polymer chain growth, the microstructures immediately swell in the aqueous solution by the osmotic pressure difference between the inside and outside of the hydrogel. The swelling kinetics are governed by the collaborative diffusion of the polymer networks and water[32,33]. For a small process zone of ~1000 μm in thickness and ~600 μm in diameter, the characteristic timescale for swelling equilibrium is ~1 min (Supplementary Fig. 11). Since the bond breaking finishes in the loading process, and the radical polymerisation almost finishes in several seconds, the swelling process in minutes is the rate-limiting process for the height change. This is further confirmed by the time-lapse microscope observation, which reveals that the microstructure does not change when observed 4 min after the mechanical indentation (Supplementary Fig. 12).

To establish a quantitative correlation between the topological feature of microstructure grown on the hydrogel surface and indentation condition, we systematically changed the indenter diameter $d$ in the range of 84–866 μm (Supplementary Fig. 13) and indention depth $L_{max}$. The lower limit of $d$ is determined by our experimental apparatus. The DN hydrogels, after being indented in monomer solutions, were

immersed in deionized water, and the height ($H$) and diameter ($D$) of the surface microstructure was determined using a three-dimensional laser microscope (Fig. 2b). The $H$ increases with the indentation depth $L_{max}$ (Fig. 2c), and the diameter $D$ hardly changes with $L_{max}$ for a fixed indenter diameter $d$. Similarly, the diameter $D$ of the microstructure almost equals that of the indenter $d$, and the height $H$ hardly changes with $d$ for a fixed $L_{max}$ (Fig. 2d, Supplementary Fig. 14). These results indicate that for the investigated experimental range, the diameter $D$ and height $H$ of the microstructure could be independently controlled by the indenter diameter $d$ and the indentation depth $L_{max}$, respectively. According to these facts, we can use force stamps with embossed micro-patterns to grow microstructures for on-demand applications.

It should be noted that the large indentation depth ($L_{max}$ = 1000 μm) also causes internal fracture of DN hydrogel with the soft surface layer (i.e. the DN hydrogel prepared on glass mould) and thereby force-triggered polymerisation beneath the surface layer. For such DN hydrogel, local swelling is also observed, whereas the surface chemical properties do not change since there is no new polymer formation in the soft layer (Supplementary Figs. 15, 16). This result confirms that the chemical modification of the topmost surface through polymerisation requires the removal of the soft surface layer. In other words, by using DN hydrogels with or without the soft surface layer, it is possible to modify the surface physical/chemical microstructure selectively.

As force-triggered radical polymerisation is generic, we further applied this method to a series of monomers (Fig. 3a). First, we measured the monomer conversion ratio of force-triggered radical

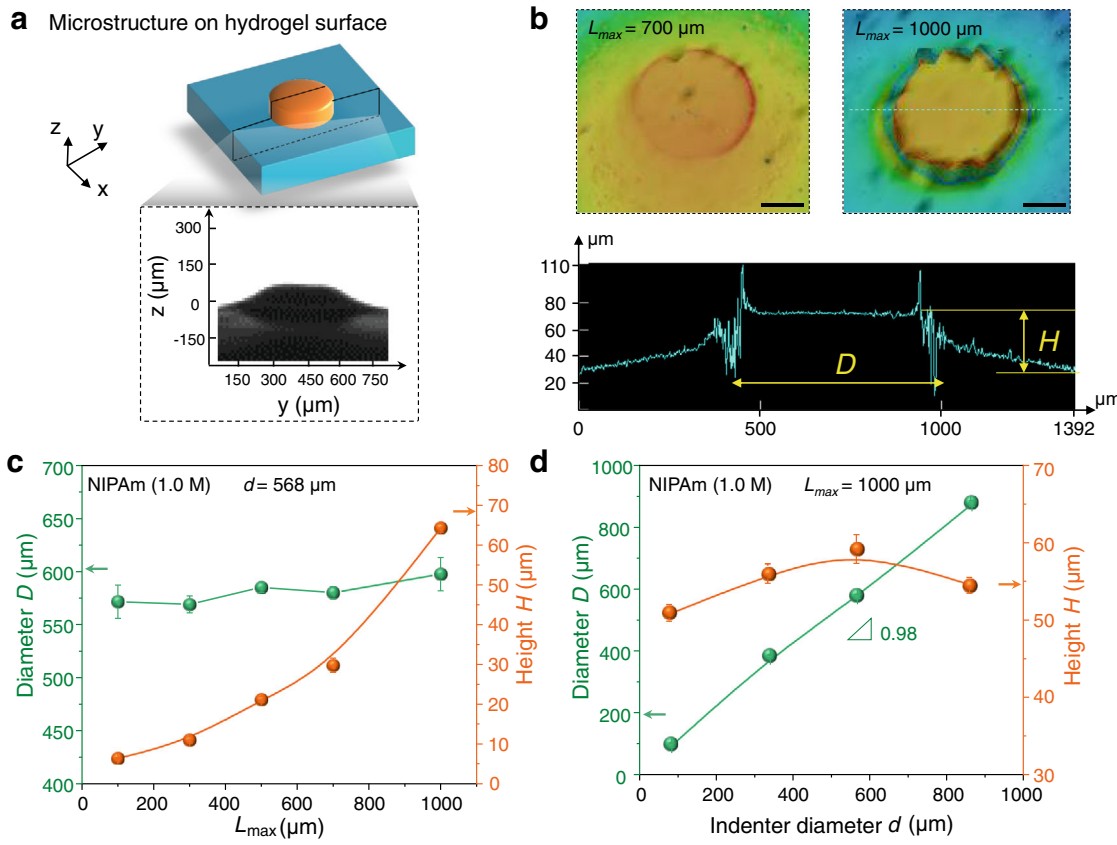

**Fig. 2 | Indenter size and depth-controlled microstructure formation.**
**a** Schematic illustration and representative microscope image (side view) of the surface microstructure on DN hydrogel after indentation by a cylindrical micro-indenter. **b** Three-dimensional laser microscope images of the surface micro-structures obtained from different indentation depths $L_{max}$ by an indenter of diameter $d$ = 568 μm, and the height profile at the position shown by the dashed line. Scale bars, 200 μm. **c** Dependence of the topographical height $H$ and diameter $D$ of

microstructures on the indentation depth $L_{max}$ by an indenter of diameter $d$ = 568 μm. **d** Topographical diameter $D$ and height $H$ of microstructures versus indenter diameter $d$ at $L_{max}$ = 1000 μm. NIPAm monomer concentration of 1.0 M was used for **a**–**d**. The observation was performed after reswelling in pure water at 25 °C. Error bars represent the standard deviation from three replicates. Some error bars in **c** and **d** are hidden by the symbols. Data in **c** and **d** are presented as mean values ± SD.

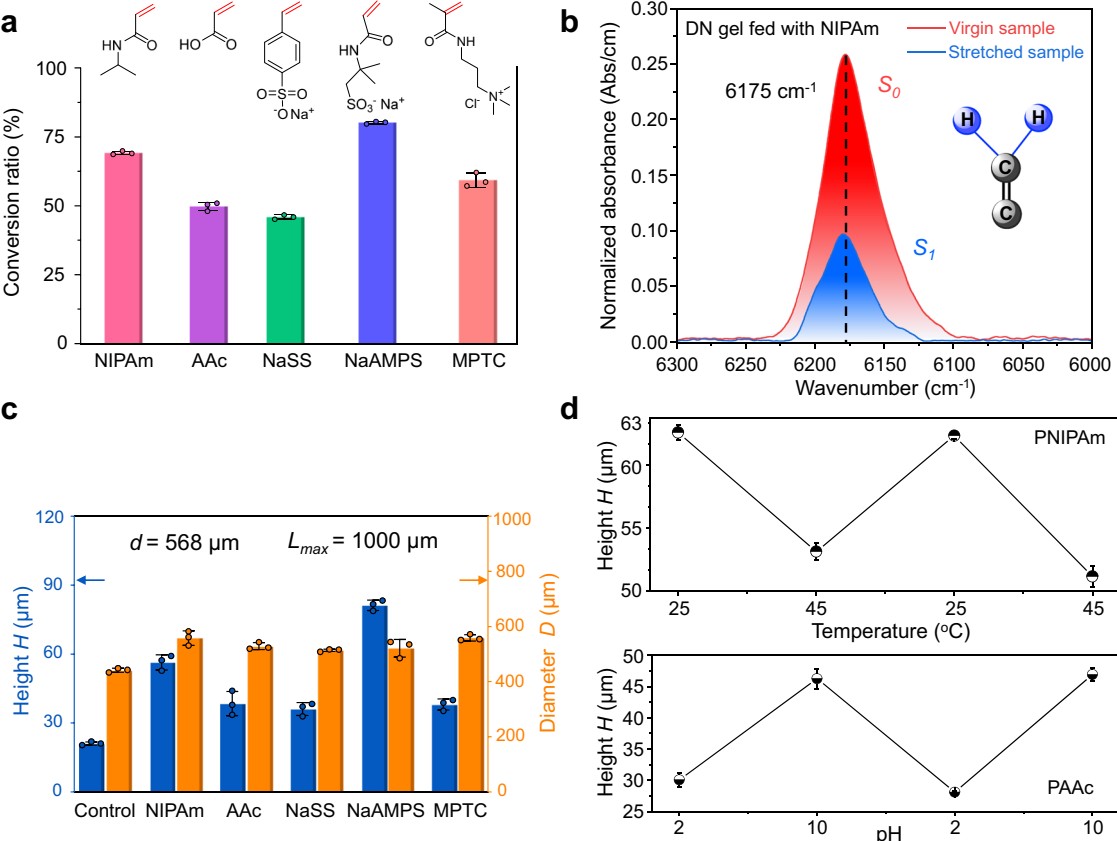

**Fig. 3 | Force-triggered microstructure growth of various polymers.**
**a** Conversion ratio of different functional unsaturated monomers in the mechanical force-triggered radical polymerisation, which was characterised in tensile specimens fed with different monomers and then stretched. The insets in **a** show the molecular structures of functional unsaturated monomers used for microstructures growth. **b** Transmission near infrared spectra of DN hydrogel fed with 1.0 M NIPAm monomer before and after uniaxial stretching to a strain $\varepsilon = 3$ (over yielding). The conversion ratio $\eta$ of monomers in **a** was calculated from peak area before ($S_O$) and after ($S_1$) stretching using $\eta = (1 - S_1/S_O) \times 100\%$. Inset, cartoon representation for the methylene with a double bond of monomer. **c** Topographical height and diameter of the microstructures for DN hydrogels indented in different monomers. Control means the DN hydrogel indented in pure water. The experiments in **a**–**c** were performed at 25 °C. **d** Cyclic changes in feature height of microstructures on DN hydrogel surfaces in response to temperature for PNIPAm grown surface (upper) and to pH for polyacrylic acid (PAAc) grown surface (lower). Error bars represent the standard deviation from three replicates. The circular dots of the histogram in **a** and **c** represent the measured data points of three. Data in **a**, **c** and **d** are presented as mean values ± SD.

polymerisation by comparing the change of monomer concentration in DN hydrogels before and after stretching of the DN hydrogels (Supplementary Figs. 17, 18). As a typical example, Fig. 3b shows the transmission near-infra-red spectra of a DN hydrogel fed with NIPAm before stretching and after stretching. From the integrated area under the spectra curves after ($S_1$) and before ($S_O$) stretching of the DN hydrogel, the conversion ratio of polymerisation $\eta = (1 - S_1/S_O) \times 100\%$ is estimated. The results reveal that the conversion ratio of NaAMPS (~80%) is the highest, and the conversion ratio of NIPAm (~69%) is higher than those of the other three monomers (acrylic acid (AAc), sodium *p*-styrenesulfonate (NaSS), and 3-(methacryloylamino)propyl-trimethylammonium chloride (MPTC)) (Fig. 3a). The difference in the conversion ratio of these reactive monomers may be related to the substituent groups on the vinyl and/or acryloyl units, or the polarity of the monomer functional groups, which determines the relative stability of the C = C bonds in the radical polymerisation[34,35].

Next, we used these monomers to fabricate various physical and/ or chemical microstructures on DN hydrogel surfaces. The typical profiles, topographic height, and diameter of the microstructures prepared in various monomer solutions by an indenter of $d = 568\,\mu\text{m}$ and $L_{max} = 1000\,\mu\text{m}$ are shown (Fig. 3c and Supplementary Fig. 19). We found that the diameters $D$ of microstructures are almost the same with the indenter diameter $d$, while the heights $H$ depend on monomer species. The monomer NaAMPS shows the highest $H$ among the

evaluated monomers. The large height of microstructure formed in NaAMPS and NIPAm monomer solutions might partly be related to their relatively high conversion ratio to polymers. A detailed discussion of the reasons is beyond the scope of this work. Furthermore, we found that adding a crosslinker *N,N'*-methylenebisacrylamide (MBA) in the monomer solution does not apparently affect both the conversion ratio and topographic height $H$ (Supplementary Figs. 20, 21a, and 22a), but $H$ increased with monomer concentration and saturated to ~60 μm in the presence of MBA (Supplementary Figs. 21b, 22b).

Depending on the functional polymers formed, these microstructures are stimuli-responsive. Specifically, cyclic temperature and pH changes bring about 30–50% reversible change in the height of micro-features formed in NIPAm and AAc, respectively (Fig. 3d, Supplementary Fig. 23). Since the bulk DN hydrogel is insensitive to heat or pH change, the induced surface microstructures could be controlled by these stimuli without altering bulk properties.

## Complex surface microstructures
Next, we created a series of complex microstructures on DN hydrogels using the force stamp method. We grew the thermo-responsive PNIPAm and used the fluorescent molecular probe (ANS) to image the complex structures grown on hydrogel surfaces. A series of stamps with different embossed patterns were prepared by using a three-dimensional (3D) printer (Supplementary Fig. 24). Figure 4a shows various PNIPAm

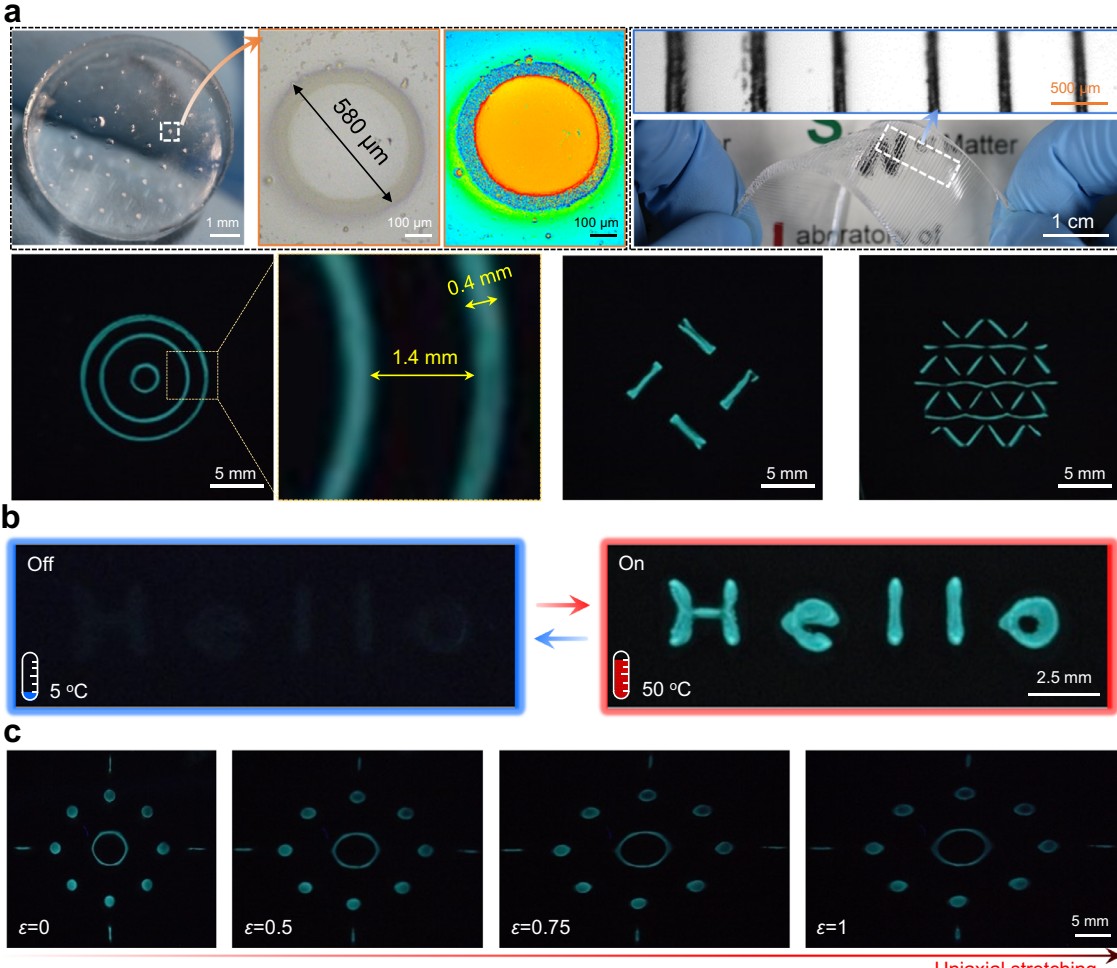

**Fig. 4 | Programmable microstructures growth on hydrogel surfaces. a** Optical and fluorescence images showing various complex structures on DN hydrogel surfaces formed by surface force-triggered growth. **b** Fluorescence images to show on-off switching of the thermal-responsive patterns to temperature change. **c** Fluorescence images to show a patterned DN hydrogel under uniaxial stretch to different strains $\varepsilon$. NIPAm 1.0 M was used for **a**–**c**.

patterns grown on DN hydrogel surfaces. The sharp boundary of the fluorescent line in Fig. 4a demonstrates fine spatial resolution of surface microstructures. In response to temperature stimuli, a dynamic fluorescent pattern that reads "Hello" was reversibly turned on/off over multiple cycles under ultraviolet (UV) light (Fig. 4b). The surface microstructures are deformable upon stretching of DN hydrogels (Fig. 4c).

**Specific applications of hydrogels with surface microstructures**
Here, we show two application examples of the hydrogels with force-triggered growth of surface microstructures. First, we demonstrate that the surface microstructures prepared by this study present chemical cues to the culture of cells. It is known that cells have a stronger adhesion to relatively hydrophobic surfaces such as biocompatible PNIPAm and poly($N,N$-dimethylacrylamide) (PDMA)[36–38]. Here, we prepared DN hydrogels with micro-patterns of PNIPAm and PDMA (Supplementary Fig. 25). A DN hydrogel damaged in deionized water, without monomer supply, was also prepared as the control sample. Myoblast cells (C2C12) labelled with a green fluorescent protein (GFP) were cultured on these DN hydrogel surfaces at 37 °C, which is higher than the lower critical solution temperature (32 °C) of PNIPAm. By day 5, on the DN hydrogels with micro-patterns of PNIPAm and PDMA, many cells were found to adhere preferentially and align on the regions with functionalized micro-patterns (Fig. 5a, Supplementary Fig. 26). A

close observation further reveals that the cells on the micro-patterns exhibit elongated morphology along with the stripe-like patterns than those on the non-patterned regions that show spherical shapes. In contrast, cells cultured on the control sample are randomly distributed, insensitive to the presence of damaged stripe patterns without growth of the functional polymers (Fig. 5b). These results reveal that the preferential growth and elongation of the cells on the micro-patterns are related to the presence of the hydrophobic polymers, not due to the surface topological effect brought by local internal damage of the DN hydrogel.

Next, we show that the anisotropic DN hydrogel surfaces with parallel PNIPAm patterns can be used for regulating surface wettability (Supplementary Fig. 27) and water droplet directional transportation (Fig. 6). When a droplet is placed on a vertically placed surface, whether it stays stationary or moves depends on the competition of wetting asymmetry of the droplet on the surface and gravity on the droplet. The resistance against sliding increases with the difference between the receding angle ($\theta_{Res}$) and advancing angle ($\theta_{Adv}$) of the droplet on a solid surface, while the driving force (i.e. gravity) for droplet sliding, increases with the droplet size. As shown in Fig. 6a–c, the difference between the receding angle and advancing angle of a water droplet (10 μL) on the surface of DN hydrogel with parallel PNIPAm lines aligned in the vertical direction is 15°, which is only half of that on the surface of DN hydrogel with PNIPAm lines aligned in the horizontal direction or on a smooth DN

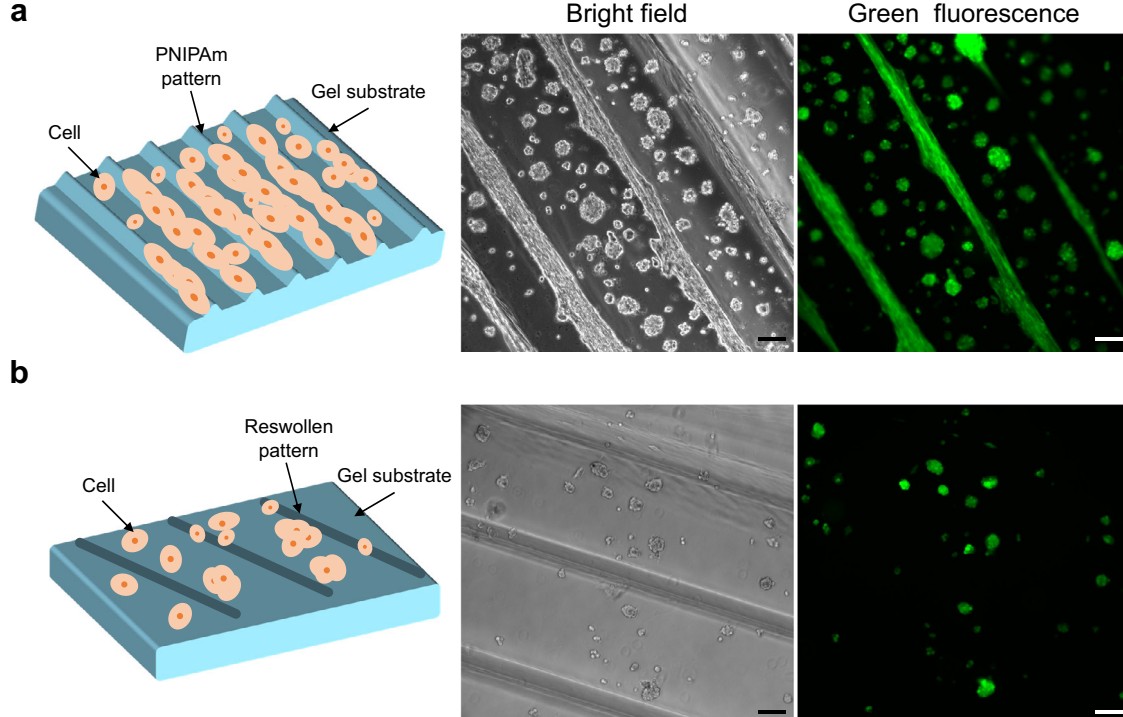

**Fig. 5 | An example to show biological application of PNIPAm-patterned DN hydrogel surfaces. a, b** Illustration, bright field and green fluorescence microscope images of cells cultured on the surface of DN hydrogel with PNIPAm strip pattern (**a**) and on the surface of DN hydrogel with internal damaged strip pattern made in water without PNIPAm (**b**). Preferential and elongated growth of cell population on the lines with PNIPAm were observed in **a** while cells distributed randomly in **b**. Myoblast cells labelled with a green fluorescence protein (GFP) were used, and the observation was performed for samples cultured at 37 °C for 5 days. Scale bars, 100 μm.

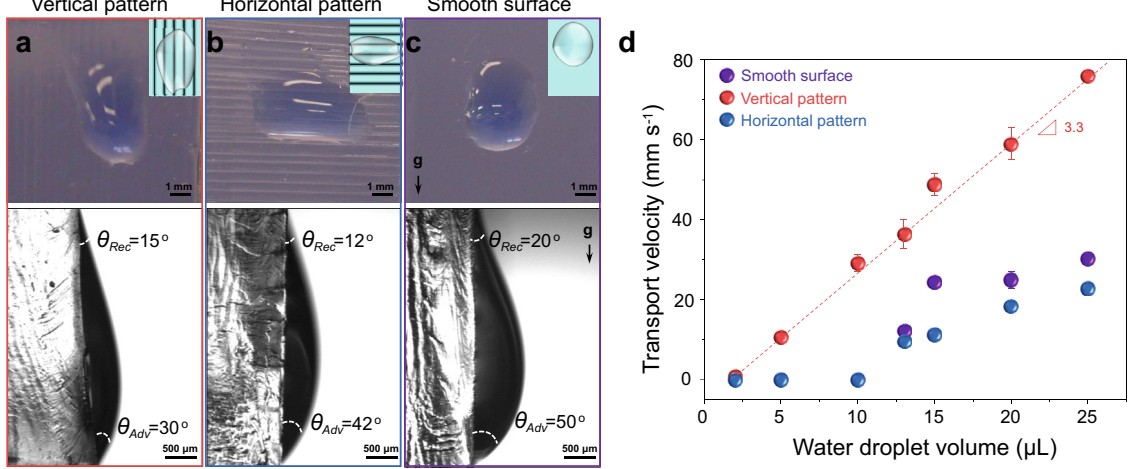

**Fig. 6 | An example to show controlled water transportation on DN hydrogels with strip-like PNIPAm patterns. a–c** Optical images to show micropattern-dependent morphologies and contact angle hysteresis of water droplets (10 μL) on DN hydrogels standing vertically. The advancing angle $\theta_{Adv}$ and receding angle $\theta_{Rec}$ are shown. The letter "**g**" and arrow indicate the direction of gravity for the three samples shown. **d** Dependence of transport velocity on water droplet volume for different hydrogel surfaces, there was a critical value around 10 μL for the smooth surface and horizontal pattern. However, the transport velocity was a linear function of water droplet volume for the vertical pattern (Slope, 3.3 mm s$^{-1}$ μL$^{-1}$). NIPAm 1.0 M was used for **a** and **b**. Error bars represent the standard deviation from three replicates. Some error bars in **d** are hidden by the symbols. Data in **d** are presented as mean values ± SD.

hydrogel surface. This result indicates that the micro-patterns can be used to regulate surface droplet transportation. In fact, the transport velocity is a linear function of water droplet volume on the patterned surface when the PNIPAm lines are set in the vertical direction. However, there is a critical value of around 10 μL for the droplet to move when the lines are set in the horizontal direction, as like on the smooth surface (Fig. 6d, Supplementary Figs. 28, 29). The experimental results demonstrate that the vertically patterned surface could induce faster water transport, and the transport velocity depends on the water droplet volume.

## Discussion

In summary, we demonstrated a facile method for rapid surface patterning of hydrogels via force-triggered growth. Such a force-triggered growth strategy is spatially controllable, allowing fine modulation of the size and shape of microstructures. Various chemical functions

could be imparted to the microstructures by using functional monomers. The on-demand micro-patterns with a variety of geometries and chemistries show promising biomimetic applications. It is worth mentioning that, unlike conventional light-triggered growth that is limited to the photoactive substrates, this force-triggered growth is, in principle, not limited to DN hydrogels but could be applied to different types of multiple-network polymeric materials. We thus envision that our rapid patterning strategy and micropatterned tough DN hydrogels hold promise for microsensor arrays, soft adhesion, flexible displays, and biomedical devices.

## Methods

### Preparation of DN hydrogels with controlled surfaces

We adopted the previously-developed technology[22] to prepare DN hydrogels with no soft second network surface layer (Supplementary Fig. 1). First, the synthesised PNaAMPS SN hydrogel (monomer concentration of 1.0 M, 40 mM crosslinker MBA, and 10 mM 2-oxoglutaric acid) was immersed in the precursor aqueous solution of the PAAm second network (monomer concentration of 2.0 M, 0.2 mM MBA, and 0.2 mM 2-oxoglutaric acid) for 1 day. Then, the immersed PNaAMPS SN hydrogel was sandwiched between two flat plates to obtain a reaction cell. The plates were made of glass covered with a silicone-coated PET film. The reaction cell was compressed at a pressure of ~0.8 kPa for 12 h. Using this method, contact of the PNaAMPS first network with the hydrophobic surface of the cell wall was ensured. The reaction cell was then irradiated with UV light (365 nm, 4 mW cm$^{-2}$) for at least 8 h in an argon glove box to synthesise the PAAm network in the presence of the first pre-stretched rigid PNaAMPS network. For reference, PAAm SN hydrogel was also synthesised from an aqueous solution of 2.0 M AAm, 0.2 mM MBA, and 0.2 mM 2-oxoglutaric acid. The synthesised DN hydrogels and reference PNaAMPS and PAAm SN hydrogels were immersed in deionised water for later use. The thickness of DN samples was 1.5–3.8 mm.

### Patterning of DN hydrogel surfaces

Rapid patterning was performed in an argon glove box using a mechanical tester (MCT-2150, A&D Co., Ltd.). First, DN samples of the required size were cut from the large pieces by a mechanical cutter and the mechanoradicals around the cutting edges are quenched prior to the use by immersing the cut samples in deionised water for two weeks. Then the DN hydrogels were re-immersed in a solution of different reactive monomers and crosslinkers overnight to incorporate the nutrients for subsequent patterning. Then, every DN hydrogel containing monomers and crosslinkers was moved to an argon glove box to remove oxygen. The DN hydrogels were fixed on a rigid glass plate with glue and then selectively indented in the solution using a cylindrical steel indenter, which was fixed on a small tensile machine using a screw. After the indentation, the gel was left still in the monomer solution for ~4 min, then it was picked up from the glove box. Finally, the hydrogels were immersed in deionised water for at least 1 min to obtain swollen microstructures on the hydrogel surfaces. The complex patterns on the DN hydrogel surfaces were prepared using 3D-printed resin stamps. The parallel patterns were prepared using stacked metal blades, the distance between which was controlled using silicone rubber spacers (thickness of ~500 μm).

### Cell adhesion to micropatterned DN hydrogel surfaces

The mouse myoblast cell line C2C12, labelled with GFP, was purchased from RIKEN Cell Bank. The cells were cultured in Dulbecco's modified Eagle's medium containing 10% foetal bovine serum and 1% penicillin/streptomycin at 37 °C in a humidified atmosphere containing 5% $CO_2$. The 4–7th passages of the C2C12 cells were used in this study. After steam sterilisation, the patterned DN hydrogel disk was placed in a six-well polystyrene tissue culture dish, and the

C2C12 cells were seeded on the PNIPAm and PDMA-patterned DN hydrogel surfaces at an initial cell density of $1 \times 10^5$ mL$^{-1}$. A cell dispersion solution was applied onto the hydrogel surface in a dropwise manner to produce a relatively uniform distribution of cells on the hydrogel surface. All cell-loaded hydrogel samples were cultured at 37 °C in a humidified atmosphere containing 5% $CO_2$. The samples were transferred to new wells for observation under a microscope. The cell morphology and proliferation on the patterned DN hydrogel surface were monitored using an Olympus IX 71 fluorescence microscope (Olympus, Japan) equipped with a digital CCD camera with a lens (magnification of 10 and 20).

### Water droplet directional transport

Droplets of deionised water of different volumes (2–25 μL) were used. The water droplets were dyed with methylene blue before use. The dyed water droplets were deposited using a micropipette. As soon as each droplet contacted the hydrogel surface, the droplet dynamics on the hydrogel surface were monitored using a commercially available camera. The relative humidity was maintained at ~30%, and the room temperature was maintained at 25 °C.

### Statistics and reproducibility

We state that at least three times, each experiment was repeated independently with similar results in our study.

Further details of these methods are available in the Supplementary Information.

### Reporting summary

Further information on research design is available in the Nature Research Reporting Summary linked to this article.

## Data availability

The authors declare that data supporting the findings of this study are available within the paper and its supplementary information files. Source data are provided with this paper.

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

## Acknowledgements

We thank decreased Dr. Daniel R. King for his contribution in editing and discussing the manuscript. We thank Y. Katsuyama for the pre-paration of the experimental devices and S. Kohsaka (National Can-cer Center Research Institute, Tokyo, Japan) for providing the C2C12-GFP cells. This research was supported by JSPS KAKENHI grants JP17H06144, JP17J09290, JP17H04891, and JP22H04968. Q.M. thanks the China Scholarship Council (CSC no. 201808120092) for financial support.

## Author contributions

Q.M., K.C., T.N., and J.P.G. conceived the concept; Q.M., K.C., T.M., Z.W., H.K., S.N., T.Y., M.F., M.T., and S.T. designed the experiments; Q.M., Z.W., H.K., S.N., T.N., M.T., S.T., and T.N. performed the experiments and analysed the data; Q.M., T.N., and J.P.G. wrote the manuscript with the help from K.C., T.M., W.C., and M.T.

## Competing interests

The authors declare no competing interests.
