## [Peer Review File · Nature Communications]

Force-triggered rapid microstructure growth on hydrogel surface for on-demand functionsReviewers' Comments:

Reviewer #1:

Remarks to the Author:

I very much enjoyed this paper, which builds off a landmark paper by many of these authors (ref. 19). The use of mechanical force to activate chemical bond formation (polymerization) is quite interesting and is nicely distinct from other forms of lithography, many of which are not suitable to hydrogel patterning. The result has reasonable generality, as demonstrated by the authors here. The level of control and resolution is sufficient to drive some applications, in particular the cell patterning and water transport examples provided.

While the applications are nice, I think the greatest impact of the paper lies in the core principle, which basically takes a perceived weakness of hydrogels (their mechanical fracture) and tames it to make it useful. I think this concept and its extension will be inspiring to materials scientists, polymer chemists, biomedical engineers, and mechanical engineers who might think of coupling instabilities to lithography to create other patterns. The writing was quite easy for me to follow and seems suitable to Nat. Commun. Finally, the videos are quite compelling (direct links from the online MS would be a nice touch).

I endorse publication following one scientific adjustment and a few edits for clarity:

Science -- the authors make a point of emphasizing the reduction in "second network outer layer" for successful surface patterning. but the process zones used to initiate response are often hundreds of microns (per the authors) and the surface layer only 1-10 microns thick. does the use of a hydrophobic mold and reduced surface layer really matter? I assume so, and that the authors have evidence of at least some isolated issues faced when using the conventional preparation methods. It would be useful to share an example or examples of cases where the surface layer has an actual impact, or to state explicitly that it is only a hypothesis that the surface layer would have a detectable impact on the technique.

Grammatical corrections --

Page 3. "Furthermore, the top-most surface of a conventional DN hydrogel is covered by a layer of the soft second network with 1–10 μm -thick. " should be "Furthermore, the top-most surface of a conventional DN hydrogel is covered by a layer of the soft second network [that is approximately] 1–10 μm -thick."

Page 4. "The internal fracture of the brittle network is proportional to the macroscopic deformation and spatially controllable and the concentration of mechanoradicals..." could be "The internal fracture of the brittle network is [both] proportional to the macroscopic deformation and spatially [controllable. The] concentration of mechanoradicals..."

"and of its second network is polyacrylamide (PAAm)." should be "and [whose] second network is polyacrylamide (PAAm)."

"...covered by a nonionic second network layer of several micrometers thick..." should be "...covered by a nonionic second network layer [that is] several micrometers thick..."

Page 6. "by covalent bonds rupture of the first network" could be "by the rupture of covalent bonds in the first network"

"the mechanoradicals concentration " should be "the [mechanoradical] concentration"

Reviewer #2:

Remarks to the Author:

In this paper, Nakajima and Gong et al demonstrated the force-induced microstructure growth on the surface of double-network hydrogels. They successfully realized the homogeneous bond breaking on the surface of double-network hydrogels to afford mechano-radicals that can polymerize free monomers in the gels. The spatially controllable structural change can bring drastic changes in surface chemistry and topology for functional change. Actually, oriented cell growth and directional water transportation were successfully demonstrated. I think the present results will contribute to the progress of mechanochemistry, polymer chemistry, surface chemistry, and materials science. The importance of this paper is high enough to warrant publication in Nature Communications, but the revision of the following minor points is required before publication.

- 1) Using a hydrophobic mold has created a double network structure on the top-most surface layer. When a hydrophilic mold is used, does the polymerization also occur in gels? Several micrometers are covered by a second network layer, whereas the indentation scale is 1000 micrometers. In that case, do the surface properties remain the same?
- 2) Why is the polymerization rate of NIPAM and NaAMPS significantly different in Supplementary Figure S6? Are the stability of the growing radicals or the diffusion rate of the monomers affected?
3. Since the NIPAM showing LCST is polymerized in Figures 2, 3, and S20, the temperature should be shown.
4. In Figure 3c, only NIPAM is non-ionic. Even if the degree of polymerization is the same, the degree of swelling may be lower.
5. Line 258: "bulk DN hydrogel is composed of thermal and pH-insensitive polymer,"
Does protonation of NaAMPS units affect the properties of DN gels?
6. PNIPAM should be able to detach cells depending on temperature, but does this system detach?

Response to Reviewers

Response to Reviewer #1

I very much enjoyed this paper, which builds off a landmark paper by many of these authors (ref. 19). The use of mechanical force to activate chemical bond formation (polymerization) is quite interesting and is nicely distinct from other forms of lithography, many of which are not suitable to hydrogel patterning. The result has reasonable generality, as demonstrated by the authors here. The level of control and resolution is sufficient to drive some applications, in particular the cell patterning and water transport examples provided.

While the applications are nice, I think the greatest impact of the paper lies in the core principle, which basically takes a perceived weakness of hydrogels (their mechanical fracture) and tames it to make it useful. I think this concept and its extension will be inspiring to materials scientists, polymer chemists, biomedical engineers, and mechanical engineers who might think of coupling instabilities to lithography to create other patterns. The writing was quite easy for me to follow and seems suitable to Nat. Commun. Finally, the videos are quite compelling (direct links from the online MS would be a nice touch).

Answer: Dear reviewer, we sincerely thank you for your time and effort in reviewing our paper. We have revised the manuscript after carefully considering your comments and suggestions. All modifications are shown in red in the revised manuscript.

I endorse publication following one scientific adjustment and a few edits for clarity:

1. Science -- the authors make a point of emphasizing the reduction in "second network outer layer" for successful surface patterning. but the process zones used to initiate response are often hundreds of microns (per the authors) and the surface layer only 1-10 microns thick. does the use of a hydrophobic mold and reduced surface layer really

matter? I assume so, and that the authors have evidence of at least some isolated issues faced when using the conventional preparation methods. It would be useful to share an example or examples of cases where the surface layer has an actual impact, or to state explicitly that it is only a hypothesis that the surface layer would have a detectable impact on the technique.

Answer: Thanks for your precious comments. Yes, we should make this important point clearer. As you commented, since our micro indentation causes internal fracture deeper than the surface layer, radical polymerization occurs in the damaged region in response to mechanical indentation regardless of the presence or absence of a surface layer. However, in the presence of a surface layer, no mechanoradical polymerization occurs on the soft surface layer. Therefore, to have the chemical modification of the hydrogel surface, it is required to use DN hydrogels without the soft surface layer.

To justify this argument, we added a result for a DN hydrogel with a soft surface layer. We performed the indentation of the DN hydrogel immersed in NIPAm solution. We confirmed the absence of surface chemical modification but the presence of polymerization in the region beneath the surface layer. We added this result in Supplementary Figures 15 and 16, and more explanation in the revised manuscript (page 8, lines 224-233).

2. Grammatical corrections --

Page 3. "Furthermore, the top-most surface of a conventional DN hydrogel is covered by a layer of the soft second network with 1–10 μm -thick. " should be "Furthermore, the top-most surface of a conventional DN hydrogel is covered by a layer of the soft second network [that is approximately] 1–10 μm -thick."

Page 4. "The internal fracture of the brittle network is proportional to the macroscopic deformation and spatially controllable and the concentration of mechanoradicals..." could be "The internal fracture of the brittle network is [both] proportional to the

macroscopic deformation and spatially [controllable. The] concentration of mechanoradicals..."

"and of its second network is polyacrylamide (PAAm)." should be "and [whose] second network is polyacrylamide (PAAm)."

"...covered by a nonionic second network layer of several micrometers thick..." should be "...covered by a nonionic second network layer [that is] several micrometers thick..."

Page 6. "by covalent bonds rupture of the first network" could be "by the rupture of covalent bonds in the first network"

"the mechanoradicals concentration " should be "the [mechanoradical] concentration"

Answer: Thanks for your helpful advice. We corrected the grammatical errors in page 3, 4, 6 in accordance with your suggestions.

Response to Reviewer #2

In this paper, Nakajima and Gong et al demonstrated the force-induced microstructure growth on the surface of double-network hydrogels. They successfully realized the homogeneous bond breaking on the surface of double-network hydrogels to afford mechano-radicals that can polymerize free monomers in the gels. The spatially controllable structural change can bring drastic changes in surface chemistry and topology for functional change. Actually, oriented cell growth and directional water transportation were successfully demonstrated. I think the present results will contribute to the progress of mechanochemistry, polymer chemistry, surface chemistry, and materials science. The importance of this paper is high enough to warrant publication in Nature Communications, but the revision of the following minor points is required before publication.

Answer: Dear reviewer, we sincerely thank you for your time and effort in reviewing

our paper. We have revised the manuscript after carefully considering your comments and suggestions. All modifications are shown in red in the revised manuscript.

1. Using a hydrophobic mold has created a double network structure on the top-most surface layer. When a hydrophilic mold is used, does the polymerization also occur in gels? Several micrometers are covered by a second network layer, whereas the indentation scale is 1000 micrometers. In that case, do the surface properties remain the same?

Answer: Thanks for your precious comments. Yes, we should make this important point clearer. As you commented, since our micro indentation causes internal fracture deeper than the surface layer, radical polymerization occurs in the damaged region in response to mechanical indentation regardless of the presence or absence of a surface layer. However, in the presence of a surface layer, no mechanoradical polymerization occurs on the soft surface layer. Therefore, to have the chemical modification of the hydrogel surface, it is required to use DN hydrogels without the soft surface layer.

To justify this argument, we added a result for a DN hydrogel with a soft surface layer. We performed the indentation of the DN hydrogel immersed in NIPAm solution. We confirmed the absence of surface chemical modification but the presence of polymerization in the region beneath the surface layer. We added this result in Supplementary Figures 15 and 16, and more explanation in the revised manuscript (page 8, lines 224-233).

2. Why is the polymerization rate of NIPAM and NaAMPS significantly different in Supplementary Figure S6? Are the stability of the growing radicals or the diffusion rate of the monomers affected?

Answer: Thanks for your comments. The difference in polymerization rate is an interesting phenomenon (shown in Supplementary Figure 6). The polymerization rate of monomers in DN hydrogels can be affected by various factors, such as the diffusion

rate of monomers and growing chains, radical stability, the interaction between the monomers/growing chains with the hydrogel networks, etc. Currently, we do not have enough experimental evidence to make clear conclusions. We would like to further investigate how the polymerization rate is related to these factors in future work.

3. Since the NIPAM showing LCST is polymerized in Figures 2, 3, and S20, the temperature should be shown.

Answer: Thanks for your comments. We have added the temperature information in Figures 2, 3, and S20.

4. In Figure 3c, only NIPAM is non-ionic. Even if the degree of polymerization is the same, the degree of swelling may be lower.

Answer: Thanks for your comments. This might be attributed to higher conversion ratio of NIPAM (69%) than that of the other three ionic monomers (acrylic acid (AAc), 50%; 3-(methacryloylamino) propyltrimethylammonium chloride (MPTC), 59%; sodium *p*-styrenesulfonate (NaSS), 46%) (Figure 3 a and c in the manuscript). The higher conversion ratio means higher polymer concentration, which results in higher osmotic pressure. For neutral polymers in semidilute region, the osmotic pressure π scales to polymer concentration Cp with a strong power law relation as $\pi \sim Cp^{2.3}$, while the osmotic pressure of polyelectrolyte linearly increases with the concentration (Polymer Physics, Oxford press). At high concentration, the neutral polymer could have osmotic pressure exceeding that of polyelectrolyte.

In the revised main text, we have modified the relevant expression as follows:

"The large height of microstructure formed in NaAMPS and NIPAm monomer solution might partly be related to their relatively high conversion ratio to polymers. "

5. Line 258: "bulk DN hydrogel is composed of thermal and pH-insensitive polymer,"
Does protonation of NaAMPS units affect the properties of DN gels?

Answer: Thanks for your comments. The protonation of NaAMPS units does not affect the properties of DN hydrogels.

We agree that NaAMPS units can be partly protonated in acidic solution. To verify whether the volume and properties of DN hydrogels can be affected by the protonation, we immersed the DN hydrogel into acid and alkaline solutions (Figures R1, R2). The results reveal that pH has no effect on swelling ratio of the DN hydrogel, suggesting effect of protonation of NaAMPS units on the properties of DN hydrogels is negligible. It has been known that swelling ratio of DN hydrogels is dominated by the dense non-ionic PAAm network owing to its high osmotic pressure, not by the sparse PNaAMPS network (Nakajima, T. et al. *Soft Matter*, 16, 5487-5496 (2020)). Even though NaAMPS units are possibly protonated at low pH, such structure change on the PNaAMPS network does not affect the swelling and related properties of DN hydrogels because of the presence of the dense PAAm network.

The sentence pointed out by the reviewer has been changed as:

"Since the bulk DN hydrogel is insensitive to heat or pH change, the induced surface microstructures could be controlled by these stimuli without altering bulk properties."

Figure R1. a. Photographs of the DN hydrogel immersed in acid solution for 24 h. b. Photographs of the DN hydrogel immersed in alkaline solution for 24 h. The shape and volume of DN hydrogel are stable in both acid and alkaline solutions.

Figure R2. Diameter change of DN hydrogels in aqueous solutions with different pH values. Here, d_0 is the initial diameter of hydrogel, d is the diameter of the hydrogel immersed in both acid and alkaline solutions for 24 h.

6. PNIPAM should be able to detach cells depending on temperature, but does this system detach?

Answer: Yes. We have preliminary found that the C2C12 cells attach on the PNIPAM-patterned DN hydrogel at 37 °C (Figure R3 a) and automatically detach at 5 °C (Figure R3 b). Since the scope of this paper is developing a novel hydrogel surface engineering method, we will report this temperature effect in a separate paper.

Figure R3. Thermoresponsive adhesion and detachment of C2C12 cells on the PNIPAM patterned hydrogels. **a.** At 37 °C, most cells significantly adhered to the PNIPAM pattern. **b.** At 5 °C, the cell clusters detached from the PNIPAM pattern automatically. Cells were incubated at 5 °C for 5 hours prior to the microscopic observation. The dotted circles mark the detached cell clusters, the red arrows show the possible detachment paths.

Reviewers' Comments:

Reviewer #1:

Remarks to the Author:

The revision addresses my concerns and I support publication.

Reviewer #2:

Remarks to the Author:

The manuscript was appropriately revised according to the comments. I think the revised version of this paper is acceptable in Nature Communications without further change.